# The Anti-Glucocorticoid Receptor Antibody Clone 5E4: Raising Awareness of Unspecific Antibody Binding

**DOI:** 10.3390/ijms23095049

**Published:** 2022-05-02

**Authors:** Lisa Ehlers, Marieluise Kirchner, Philipp Mertins, Cindy Strehl, Frank Buttgereit, Timo Gaber

**Affiliations:** 1Department of Rheumatology and Clinical Immunology, Charité-Universitätsmedizin Berlin, Corporate Member of Freie Universität Berlin, Humboldt-Universität zu Berlin, 10117 Berlin, Germany; lisa.ehlers@charite.de (L.E.); cstrehl1102@gmail.com (C.S.); frank.buttgereit@charite.de (F.B.); 2Deutsches Rheuma-Forschungszentrum (DRFZ), Institute of the Leibniz Association, 10117 Berlin, Germany; 3BIH Core Unit Proteomics, Berlin Institute of Health (BIH) and Max-Delbrück Centrum für Molekulare Medizin (MDC), 13125 Berlin, Germany; marieluise.kirchner@mdc-berlin.de (M.K.); philipp.mertins@mdc-berlin.de (P.M.)

**Keywords:** antibody specificity, cross-reactivity, replication crisis, glucocorticoid receptor

## Abstract

Unspecific antibody binding takes a significant toll on researchers in the form of both the economic burden and the disappointed hopes of promising new therapeutic targets. Despite recent initiatives promoting antibody validation, a uniform approach addressing this issue has not yet been developed. Here, we demonstrate that the anti-glucocorticoid receptor (GR) antibody clone 5E4 predominantly targets two different proteins of approximately the same size, namely AMP deaminase 2 (AMPD2) and transcription intermediary factor 1-beta (TRIM28). This paper is intended to generate awareness of unspecific binding of well-established reagents and advocate the use of more rigorous verification methods to improve antibody quality in the future.

## 1. Introduction

Over the last century, antibodies have become indispensable, both as tools in biomedical research and as therapeutic agents. At the same time, these reagents are considered major contributors to the replication crisis. Researchers found that less than a quarter of important preclinical studies were reproducible [1,2]. These failures to validate previous findings cause yearly global costs of about USD 28 billion [3]. Defective reagents account for more than one third of this sum. Initiatives to improve antibody quality were launched to reduce this burden imposed on research resources [4,5]. Despite these advances, a single recommended approach for antibody validation has not yet been established. Here, we identify considerable unspecific binding of a commonly used monoclonal antibody that had previously been verified by a combination of generally recognized methods.

The mouse monoclonal anti-glucocorticoid receptor antibody clone 5E4 was raised against a 26 amino acid peptide (APTEK-26). This epitope represents the amino acids 150–176 of the glucocorticoid receptor (GR) and is situated in a conserved region of the regulatory part of the receptor [6]. Berki et al. initially verified the antibody’s capacity to bind its target by ELISA and western blot. Among other researchers, our group subsequently used this reagent to visualize the GR. After confirming expression of the membrane-bound GR (mGR) on primary immune cells, we examined the receptor functionally and identified differences in surface expression under inflammatory conditions with the help of the anti-GR antibody clone 5E4 [7,8,9,10,11]. In order to further characterize the mGR, we performed immunoprecipitation followed by mass spectrometry (IP-MS). These analyses uncovered the predominant pull down of two unexpected targets by this antibody clone. By describing our findings in this paper, we aim to raise awareness for unspecific antibody binding and encourage the widespread introduction of more rigorous validation procedures.

## 2. Results

### 2.1. Anti-GR (5E4) Surface Staining

As we had previously identified HEK293 cells as a model system characterized by ample mGR expression [11], we initially verified surface-staining specificity in this cell line. Flow cytometry revealed almost complete prevention of surface staining by prior incubation with excess unconjugated antibody or APTEK-26 peptide, excluding the possibility of unspecific attachment of the antibody to the cell surface (Figure 1A). This was additionally verified in different cell lines as well as primary human immune cells (Appendix A). To isolate and characterize the mGR, we established conditions that modified mGR expression, thus serving as positive and negative controls in the subsequent experiments.

The signal yielded by surface staining with the anti-GR (5E4) antibody was reduced by treating the cells with inhibitors of Golgi transport and protein palmitoylation (Figure 1B). On the other hand, immunostimulation enhanced mGR expression: THP-1 cells displayed a significant increase in anti-GR (5E4) surface-staining intensity upon incubation with PMA. These results were in line with our previous findings demonstrating an upregulation of monocytic mGR expression upon stimulation with LPS in vitro as well as under conditions of immunoactivation in vivo [7,8,9,10].

### 2.2. Reduced Anti-GR (5E4) Antibody Signal after Stable Glucocorticoid Receptor Gene Silencing

Our group had previously created a GR knockdown in HEK293 cells by RNA interference [11]. Western blotting of HEK293 whole cell lysates stably expressing shRNA directed against the GR revealed a reduction in protein expression using the anti-GR (5E4) antibody. Flow cytometric analysis verified this decrease with respect to mGR expression by surface staining of transduced HEK293 cells (Figure 1D).

### 2.3. Immunoprecipitation Using Anti-GR (5E4) Antibody

To characterize the mGR protein, we enriched GR protein from HEK293 whole cell lysates and membrane fractions by immunoprecipitation (IP) with the anti-GR (5E4) antibody. Western blot analyses of immunoprecipitates from HEK293 membrane fractions yielded a band of the expected molecular weight of approximately 100 kDa (Figure 2A). Specificity was confirmed by including an isotype control: control samples incubated with mouse IgG1 in parallel did not produce a protein band in the 100 kDa area (Figure 2B). For both the immunoprecipitation and the isotype control samples, this target area was extracted from the gel and analyzed by mass spectrometry. Surprisingly, these measurements revealed two different proteins sized about 100 kDa, namely TRIM28 and AMPD2, as the top abundant and most enriched candidates in the pull-down sample obtained by IP using the anti-GR (5E4) antibody.

To validate this finding and exclude any membrane enrichment-related differences in antibody specificity, the IP experiments were repeated using whole cell lysates. Subsequent mass spectrometric analysis confirmed TRIM28 and AMPD2 as the two most enriched proteins by pull down with the anti-GR (5E4) antibody (Figure 2C, Appendix A). Similar results were obtained with whole cell lysates from two additional cell lines—Jurkat and THP-1 cells (Figure 2C, Appendix A).

### 2.4. Verification of Anti-GR (5E4) Antibody Specificity

Following these initial observations indicating unexpected specificity of anti-GR (5E4) antibody, we aimed to explore these findings in more detail.

First, we investigated potential antibody batch effects by replicate experiments applying the antibody clone 5E4 provided by different manufacturers. All approaches yielded similar results: Western blot analyses of pull-down samples obtained by IP using different anti-GR (5E4) antibody lots demonstrated identical bands at a molecular weight of about 100 kDa (Figure 2D). Subsequent mass spectrometric analyses identified TRIM28 and AMPD2 in these enriched samples (Figure 2E, Appendix A). Of note, GR was also detected in these pull-down samples, although both intensity and enrichment were decidedly weaker. In contrast, two other independent anti-GR antibodies—mouse monoclonal anti-GR (G-5) and rabbit polyclonal anti-GR (pAb PA1)—showed high specificity for the GR, resulting in efficient GR protein enrichment from the cell lysates (Figure 2E, Appendix A).

Western blot analysis confirmed that proteins enriched by anti-GR (5E4) IP were indeed detectable by both anti-TRIM28 and anti-AMPD2 antibodies (Figure 2F).

Generally, there are three major causes of incorrect antibody binding: (i) signal interference of bait-interacting proteins (co-immunoprecipitation), (ii) contamination with a different clone, and (iii) cross-reactivity.

We excluded interference by GR protein interactors by performing IP using the specific anti-GR (G-5) antibody. Mass spectrometric analyses confirmed that TRIM28 and AMPD2 were not detected in these pull-down samples despite strong enrichment of GR protein (Figure 2E). This finding was also confirmed by western blot, although the signal of the target protein was low in anti-GR (G-5) and anti-TRIM28 pull-down samples (Figure 3A). In addition, GR abundance in 5E4 antibody-based pull-downs was significantly lower compared to TRIM28 and AMPD2, excluding the possibility of co-elution as the cause of differential specificity.

Clone contamination resulting in multiple specificities was excluded as a cause by testing different antibody batches (Figure 2D,E).

To further elucidate the specificity of the antibody clone, we performed IP with the anti-GR (5E4) antibody with and without prior incubation with APTEK-26 peptide—the antibody’s epitope. Quantitative mass spectrometric analysis revealed that the peptide pre-incubation resulted in decreased abundance of TRIM28 and AMPD2 in the pull-down samples, while the GR was only slightly reduced (Figure 3B, Appendix A). This decrease in TRIM28 and AMPD2 enrichment would not have been expected in the case of clone contamination. The most likely cause of TRIM28 and AMPD2 signals in anti-GR (5E4) antibody pull-down samples is, therefore, cross-reactivity. Due to the strong reduction of enrichment following the peptide block, we assumed that conformational homology of the epitope region might account for unspecific protein binding. By blasting the amino acid sequences of TRIM28 and AMPD2 against the APTEK-26 peptide, we did indeed observe some overlap (Figure 3C). Both alignments are situated at the beginning of the APTEK-26 peptide, which corresponds with the likely anti-GR (5E4) antibody-binding site described by Berki et al. [6].

### 2.5. Independent Validation of the Antibody Target Using the Anti-GR (G-5) Antibody

To discern the true target of the anti-GR (5E4) staining, which was crucial to our previous work on mGR expression, we re-evaluated several conditions using the anti-GR antibody clone G-5. As described above, we verified antibody specificity by IP-MS prior to the following experiments. We also ensured successful blocking of the surface staining by adding excess unconjugated antibody (Appendix A). Surface stainings for TRIM28 and AMPD2 were performed in parallel. Flow cytometry revealed that the surface-staining pattern of different cell lines obtained using the anti-GR (5E4) antibody did not correspond with any of the examined target proteins (Figure 4A). With respect to the decrease in staining intensity observed after Golgi transport inhibition, anti-GR (G-5) staining demonstrated that this finding was not caused by a reduction in mGR expression (Figure 4B). Similarly, an increase in mGR expression does not account for the rise in monocytic anti-GR (5E4) staining intensity observed after immunostimulation (Figure 4C). Staining THP-1 cells after incubation with PMA highlighted the difficulty of determining the true target of the anti-GR (5E4) surface staining most strikingly: none of the three proposed target proteins displayed the expected increase in surface expression the anti-GR (5E4) antibody had previously indicated (Figure 4D). Eventually, surface staining performed in HEK293 cells after GR knockdown underlined the importance of considering off-target effects of RNA interference: while a reduction in staining intensity after knockdown of the target protein is generally considered as an acceptable approach to validate antibody specificity, we observed a similar decrease with respect to AMPD2 surface expression.

## 3. Discussion

This study provides an example of relevant unspecific binding of a widely used commercially available monoclonal antibody. Using IP-MS, we identified TRIM28 and AMPD2 as the predominant targets of the anti-GR antibody clone, 5E4. We confirmed these results by western blot analysis and illustrated the value of independent antibody strategies to assess antibody specificity.

In recent years, researchers have increasingly published studies identifying poorly validated protein affinity reagents as a major source of inconsistent results and unsuccessful translational research [14,15,16,17]. These findings have underlined the need for adequate antibody validation and fueled initiatives to establish appropriate guidelines. However, these recommendations are not binding, and a single uniformly approved approach has not yet been defined. The question of how rigorous antibody validation should be remains unsolved. Our study highlights that complying with a substantial portion of the existing recommendations does not exclude unspecific antibody binding that can significantly compromise the results of years of careful research.

While antibody characterization certainly represents a fundamental prerequisite for appropriate antibody use, this information does not sufficiently ensure antibody quality. According to the minimum information about a protein affinity reagent (MIAPAR) proposal, users are encouraged to provide information on the following parameters to describe an affinity reagent: producer, target, production, affinity reagent identifier, affinity reagent class, host organism, epitope, technical applications as well as the procedure by which the reagent has been characterized [18]. These data defining the anti-GR antibody clone, 5E4, are fully available [6]. Similarly, SciCrunch (http://scicrunch.org/resources, accessed on 7 August 2021)—a database developed to improve characterization of research antibodies as part of the resource identification initiative [19]—provides a research resource identifier (RRID) for this reagent. 

In 2016, the International Working Group for Antibody Validation (IWGAV) published a proposal for antibody validation comprising five pillars: “(i) genetic strategies, (ii) orthogonal strategies, (iii) independent antibody strategies, (iv) expression of tagged proteins, and (v) immunocapture followed by mass spectrometry” [5]. The authors recommend that antibody validation should incorporate at least one of these categories. With respect to the anti-GR antibody clone 5E4, we fulfilled this criterion by performing GR knockdown. Additional methods that are commonly used to verify antibody specificity were successfully applied: adsorption with APTEK-26 peptide eliminated the staining signal measured by flow cytometry; IP produced a single band of the expected molecular weight; several conditions were established that modified mGR expression determined by anti-GR (5E4) antibody surface staining.

Despite these efforts, we were faced with the unwelcome discovery that the GR does not represent the predominant target of the antibody clone, 5E4. In light of these findings, we aim to raise awareness that even a set of seemingly accurate controls does not exclude the possibility of cross-reactivity. While the reduction in GR signal measured by both flow cytometry and western blot after GR knockdown indicated antibody specificity, we also observed a reduction in AMPD2 surface expression in HEK293 cells expressing the shRNA constructs. We later learned that genes coding AMP deaminases were targets of GR signaling [20,21]. Additionally, alignment of up to nine nucleotides was identified when blasting GR-targeting shRNA (Table 1) and *AMPD2* gene sequences, accounting for potential off-target effects. This possible explanation for our misleading findings also underlined the importance of recognizing off-target effects of genetic strategies. Nevertheless, we do not dismiss the value of this pillar to exclude cross-reactivity. In fact, Taves et al. recently identified a significant staining produced by the anti-GR antibody clone 5E4 in GR-deficient thymocytes [22]. This was not observed using the antibody clone G-5 whose specificity we confirmed by IP-MS analysis. Interestingly, while IP-MS identified AMPD2 as the target protein most efficiently pulled down by the anti-GR (5E4) antibody, pre-incubation with the immunizing peptide, APTEK-26, had a more pronounced effect on the IP of TRIM28. These findings suggest that the area relevant for AMPD2 binding might extend beyond the APTEK-26-binding site. Additionally, different binding affinities might play a role in this context. These characteristics were not determined as part of our work.

Our results show that independent antibody strategies also represent a useful pillar—especially when genetic strategies and IP-MS are not applicable. By performing parallel stainings with the anti-GR antibody clones 5E4 and G-5, we identified striking differences in the staining patterns produced by the two reagents. These findings would have provided an indication of the unspecific binding of either reagent without requiring additional methods, such as RNA interference or mass spectrometry.

In summary, our study promotes rigorous antibody validation by underlining the dangers of misleading results produced by less stringent methods. We consider these measures particularly relevant when the target protein has been little examined and comparable data are sparse.

## 4. Materials and Methods

### 4.1. Preparation of Peripheral Blood Mononuclear Cells and Magnetic Cell Separation

Peripheral venous blood for isolation of peripheral blood mononuclear cells (PBMCs) was collected in lithium heparin tubes. PBMCs were isolated by Ficoll-Paque^TM^ PLUS (GE Healthcare, Chicago, IL, USA) density gradient centrifugation according to the manufacturer’s instructions. Where applicable, PBMC isolation was followed by magnetic cell separation (MACS) using anti-CD14-microbeads (Miltenyi Biotec, Bergisch Gladbach, Germany) to obtain CD14+ monocytes.

### 4.2. Chemicals and Reagents

Dulbecco’s Modified Eagle Medium (DMEM), Roswell Park Memorial Institute 1640 Medium (RPMI), penicillin, streptomycin, and L-glutamine were obtained from Gibco^TM^ (Thermo Fisher Scientific, Waltham, MA, USA). Fetal calf serum (FCS), β-mercaptoethanol (2-ME), polybrene, Accutase® solution, brefeldin A (BFA), monensin (MN), 2-bromohexadecanoic acid (2-BP), lipopolysaccharide (LPS), phorbol 12-myristate 13-acetate (PMA), and dexamethasone (Dex) were purchased from Sigma-Aldrich (St. Louis, Missouri).

### 4.3. Antibodies

Staining for flow cytometry was performed using antibodies against AMPD2 (rabbit polyclonal, Thermo Fisher Scientific, Cat# PA5-26127, RRID:AB_2543627), CD14 (TM1, DRFZ, Berlin, Germany), CD16 (3G8, BioLegend, San Diego, CA, USA, Cat# 302018, RRID:AB_314218), GR (5E4, provided by Timea Berki [6]), GR (G-5, SCBT, Dallas, TX, USA, Cat# sc-393232 PE, RRID:AB_2687823), and TRIM28 (rabbit polyclonal, LSBio, Seattle, WA, USA, Cat# LS-C211592). Goat anti-rabbit IgG (Invitrogen, Carlsbad, CA, USA, Cat# P2771MP, RRID:AB_221651) and streptavidin (Life Technologies, Carlsbad, CA, USA, Cat# S866) were used as secondary reagents. Antibodies against GR (5E4, provided by Timea Berki [6]), GR (G-5, SCBT, Cat# sc-393232, RRID:AB_2687823), AMPD2 (rabbit polyclonal, Thermo Fisher Scientific, Cat# PA5-26127, RRID:AB_2543627), TRIM28 (rabbit polyclonal, Thermo Fisher Scientific, Cat# PA5-27648, RRID:AB_2545124), beta-actin (BA3R, Invitrogen, Cat# MA5-15739, RRID:AB_10979409), mouse IgG (Promega, Madison, WI, USA, Cat# W4021, RRID:AB_430834), and rabbit IgG (Promega, Cat# W4011, RRID:AB_430833) were used for western blot analysis. Streptavidin conjugated to horseradish peroxidase (HRP) (R&D Systems, Minneapolis, MN, USA, Cat# DY998) was used after application of biotinylated primary antibodies. IP was realized with the help of the following antibodies: mouse monoclonal anti-GR antibody (5E4, provided by Timea Berki [6] and Bio-Rad, Cat# MCA2469, RRID:AB_10844347), mouse monoclonal anti-GR antibody (G-5, SCBT, Cat# sc-393232, RRID:AB_2687823), rabbit polyclonal anti-GR antibody (pAb PA1) (Thermo Fisher Scientific, Cat# PA1-511A, RRID:AB_2236340), mouse monoclonal anti-AMPD2 antibody (QQ13, SCBT, Cat# sc-100504, RRID:AB_2258261), and rabbit polyclonal anti-TRIM28 antibody (Thermo Fisher Scientific, Cat# PA5-27648, RRID:AB_2545124). Mouse IgG1 (Invitrogen, Cat# 02-6100, RRID:AB_2532935), mouse IgG2b kappa (Invitrogen, Cat# 14-4732-82, RRID:AB_470117), and rabbit IgG (Invitrogen, Cat# 02- 6102, RRID:AB_2532938) served as isotype controls.

### 4.4. Cell Culture

All cell lines were purchased from ATCC® (Manassas, VA, USA). Human embryonic kidney 293 (HEK293) cells (Cat# CRL-1573, RRID:CVCL_0045) were cultured in DMEM supplemented with 10% (*v*/*v*) FCS. Jurkat (Cat# TIB-152, RRID:CVCL_0367), THP-1 (Cat# TIB-202, RRID:CVCL_0006), and CCRF-CEM (Cat# CCL-119, RRID:CVCL_0207) cells as well as human primary immune cells were cultured in RPMI supplemented with 10% (*v*/*v*) FCS. Amounts of 100 U/mL penicillin, 100 µg/mL streptomycin, and 50 µM 2-ME were added to all media used. Cells were incubated in a humidified atmosphere at 5% CO2 and approximately 18% O2. Adherent cells were detached with the help of Accutase® solution. Immunostimulation was achieved by incubation with 1 µg/mL LPS or 10 ng/mL PMA, as indicated. Inhibition of the secretory pathway was achieved by incubation with 1 µg/mL BFA or 0.5 µg/mL MN. A total of 100 µg/mL 2-BP was added to inhibit protein palmitoylation. Dex was added at a concentration of 10^−5^ M.

### 4.5. Reduction of GR Gene Expression by RNA Interference

Stable GR knockdown in HEK293 cells was established, as described previously [11]. shRNA sequences are provided in Table 1. The pLentiLox3.7 vector (Addgene plasmid #11795; http://n2t.net/addgene:11795, accessed on 13 November 2020; RRID:Addgene_11795) was used to generate shRNA constructs. AllStars Negative Control siRNA (Qiagen, Hilden, Germany) served as a negative control. Co-transfection of HEK293 cells with the lentiviral packaging plasmids pVSVG and pPAX2 by means of calcium phosphate precipitation yielded viral particles. Supernatants were collected after 48-72 h and supplemented with 8 µg/mL polybrene. HEK293 cells were infected by centrifugation for 90 min at 700× *g* and 37 °C. Successfully transduced cells were identified by expression of green fluorescent protein.

### 4.6. Sample Preparation for Protein Analysis

Whole cell lysates for IP were obtained by lysing 10 × 10^6^ cells with IP lysis buffer (10 mM Tris HCl pH 7.5, 10 mM NaCl, 2 mM EDTA, 0.1% (*v*/*v*) Triton X-100, 1 mM PMSF, 2 µg/mL aprotinin). We purified cytosolic and membrane fractions using the Mem-PER™ Plus Membrane Protein Extraction Kit (Thermo Fisher Scientific) according to the manufacturer’s protocol. Protein concentration was determined by bicinchoninic acid assay (BCA assay, Interchim, Montluçon, France). IP was performed overnight at 4 °C by incubating the lysates with the antibodies listed above. We included identical samples incubated with a corresponding isotype control in every experiment. Pull down of the antibody-target complex was achieved with the help of agarose-conjugated protein, A/G PLUS (SCBT). The samples were washed with IP buffer (0.15 M NaCl, 0.05 M Tris-HCl pH 8, 1% (*v*/*v*) NP40) at 1000× *g* and 4°C. Laemmli sample buffer (Bio-Rad, Hercules, CA, USA) was added to the agarose pellets prior to sodium dodecyl sulfate polyacrylamide gel electrophoresis (SDS-PAGE). Samples intended for mass spectrometric analysis were either extracted from the gel after identification by Pierce Coomassie Brilliant Blue G-250 Dye (Thermo Fisher Scientific) or digested directly as described below.

### 4.7. Western Blot Analysis

Protein samples were separated by SDS-PAGE and subsequently blotted onto PVDF membranes (Millipore, Burlington, MA, USA). The primary antibodies used for protein detection were applied, as indicated, and visualization was achieved through enzymatic chemiluminescence (PerkinElmer, GE Healthcare).

### 4.8. LC-MS/MS Analyses

Gel band samples from SDS-PAGE analyses were processed for mass spectrometric analyses using in-gel digestion with trypsin, as previously described [23]. Laemmli sample buffer containing samples were cleaned up and trypsin digested using SP3 protocol [24]. For all other pull-down samples, the washed beads were directly processed using on-bead tryptic digestion [25]. Peptide samples were desalted using the StageTips protocol [26], separated by reversed-phase chromatography on in-house manufactured 20 cm fritless silica microcolumns with an inner diameter of 75 µm, packed with ReproSil-Pur C18-AQ 3 µm resin (Dr. Maisch GmbH, Ammerbuch-Entringen, Germany), using a gradient (45 or 98 min) of increasing Buffer B concentration (from 2% to 60%, Buffer B: 90% acetonitrile) with a 250 nL/min flow rate on an High-Performance Liquid Chromatography (HPLC) system (Thermo Fisher Scientific). Eluting peptides were directly ionized by electrospray ionization and transferred into a Q Exactive Plus or Orbitrap Fusion mass spectrometer (Thermo Fisher Scientific). The Q Exactive Plus instrument was operated in data-dependent mode with performing full scans followed by top 10 MS2 scans. The Orbitrap Fusion instrument was operated in top speed mode with 3 s cycles. Raw data were analyzed using the MaxQuant software (v1.6.3.4), as described [27]. The internal Andromeda search engine was used to search MS2 spectra against a decoy UniProt database for mouse (HUMAN.2019-07) containing forward and reverse sequences. The search included variable modifications of oxidation (M) and N-terminal acetylation, deamidation (N and Q), and fixed modification of carbamidomethyl cysteine. Minimal peptide length was set to seven amino acids and a maximum of two missed cleavages was allowed. The false discovery rate (FDR) was set to 1% for peptide and protein identifications. Unique and razor peptides were considered for quantification. The integrated “match between runs” option as well as label-free quantification (LFQ) and intensity-based absolute quantification (iBAQ) calculation algorithm were activated. The resulting text files were filtered to exclude reverse database hits, potential contaminants (including immunoglobulin species), and proteins only identified by site. For in-gel samples, iBAQ data were filtered for minimum three peptides and two MS/MS counts. For pull-down data, LFQ values were used and the following protein filters were applied: minimum two peptides, minimum two MS/MS counts, minimum three valid in values in at least one experimental group. Missing values were imputed with low-intensity values following normal distribution. Significance cut-off for group comparison was set at FDR 5%.

### 4.9. Flow Cytometry

Staining for flow cytometry was performed on ice by applying the antibodies listed above. To block unspecific binding of Fc receptors, 10% (*v*/*v*) human IgG (Kiovig [50 mg/mL], Baxter AG, Vienna, Austria) was added. Dead cells were excluded with the help of 7-AAD (BD) or DAPI (Sigma-Aldrich). Successful blocking of the staining with 100-fold excess unconjugated antibody excluded the possibility of unspecific antibody attachment to the cell surface. The samples were measured using a MACSQuant Analyzer 10 (Miltenyi Biotec) and analyzed with FlowJo™ software (version 7.6.4, BD). Staining intensity is depicted as geometric mean fluorescence intensity (gMFI). Results are provided as the ratio (r gMFI) of staining to either block or unstained control, as indicated. The gating strategy is depicted in Appendix A.

### 4.10. Statistical Analysis

Statistical analyses were performed using GraphPad Prism. Wilcoxon matched-pairs signed-rank test was applied in order to assess differences between paired samples. For mass spectrometric analyses, differential protein abundance was calculated using the two-sample Student’s t test.

## Figures and Tables

**Figure 1 ijms-23-05049-f001:**
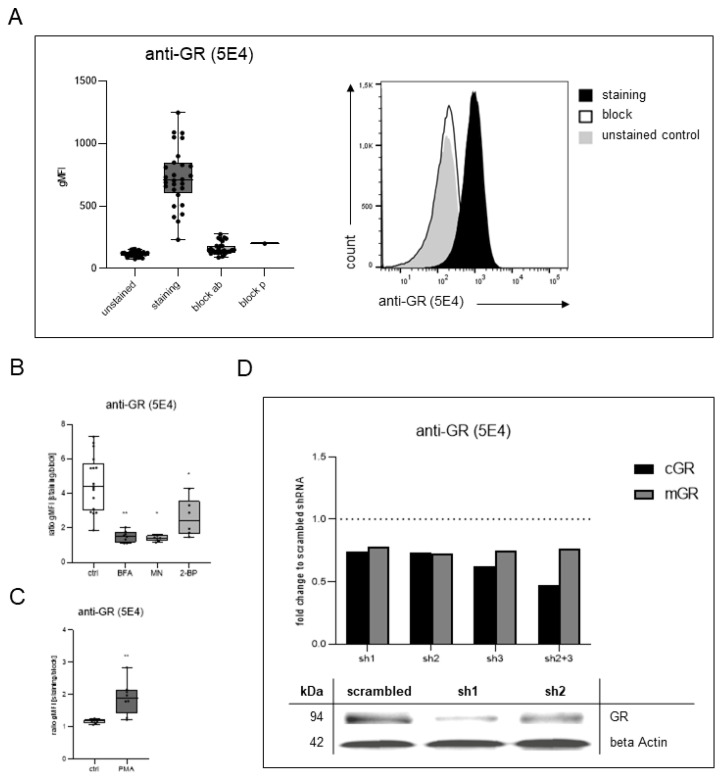
Anti-GR (5E4) antibody surface staining. (**A**) HEK293 cells were analyzed for mGR expression by flow cytometry using the anti-GR (5E4) antibody (n = 28). The staining was blocked successfully by 10-min incubation with 100-fold excess unconjugated primary antibody (block ab) or APTEK-26 peptide (block p) prior to the staining procedure. The gating strategy is displayed in Appendix A. Staining intensities are depicted as geometric mean fluorescence intensity (gMFI). (**B**) mGR expression on HEK293 cells after inhibition of Golgi transport and protein palmitoylation. Cells were incubated with 1 μg/mL BFA, 0.5 µg/mL MN, and 100 µg/mL 2-BP, respectively, for 24 h and mGR expression was measured by flow cytometry using the anti-GR (5E4) antibody (n = 6–8). The cells were gated according to Appendix A for analysis. r gMFI represents the ratio of geometric mean fluorescence intensity of staining to block with excess unconjugated antibody. (**C**) mGR expression on THP-1 cells after immunostimulation with 10 ng/mL PMA for 24 h. mGR expression was measured by flow cytometry using anti-GR (5E4) antibody (n = 8). The cells were gated according to Appendix A for analysis. r gMFI represents the ratio of geometric mean fluorescence intensity of staining to block with excess unconjugated antibody. (**D**) GR protein expression after stable GR knockdown detected by anti-GR (5E4) antibody. cGR levels were determined by western blot following SDS-PAGE. Protein levels were normalized to beta-actin and are displayed as fold-change to scrambled shRNA control samples. mGR levels were measured by flow cytometry. The data represent the ratio of geometric mean fluorescence intensity of staining to block with excess unconjugated antibody and are depicted as fold-change to scrambled shRNA control samples. Data modified from Strehl 2011 [12]. All boxplots show median, interquartile range, and minimum and maximum values, respectively. * *p* < 0.05, ** *p* < 0.01, compared to untreated control; Wilcoxon matched-pairs signed-rank test. Legend: 2-BP, 2-bromohexadecanoic acid; BFA, brefeldin A; cGR, cytosolic glucocorticoid receptor; mGR, membrane-bound glucocorticoid receptor; MN, monensin; PMA, phorbol 12-myristate 13-acetate.

**Figure 2 ijms-23-05049-f002:**
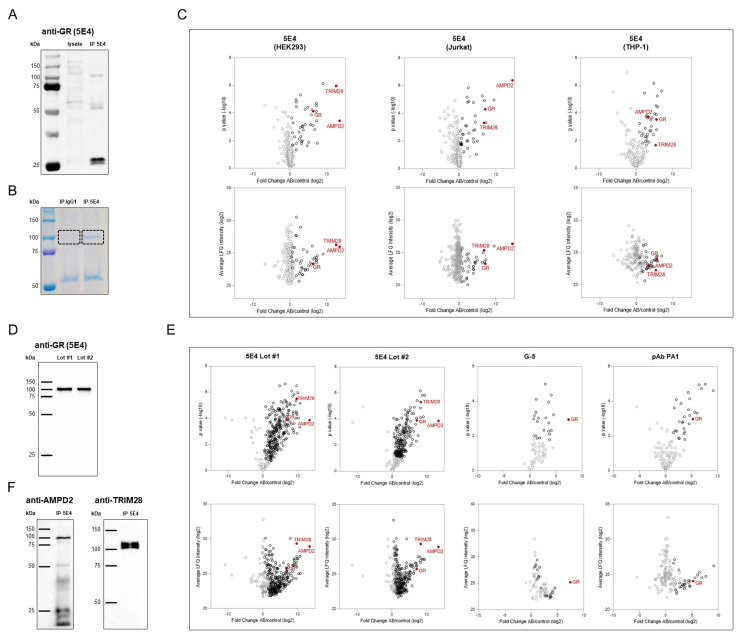
Verification of anti-GR (5E4) antibody specificity. (**A**) Western blot analysis of GR pulled down from HEK293 membrane fractions by immunoprecipitation using the anti-GR (5E4) antibody (IP 5E4). An amount of 20 µL of membrane fraction protein (lysate) were analyzed in parallel. Protein detection was achieved by adding the anti-GR (5E4) antibody followed by an HRP-conjugated anti-mouse IgG antibody as a secondary reagent. (**B**) Immunoprecipitation from HEK293 membrane fractions was performed using the anti-GR (5E4) antibody (IP 5E4) and mouse IgG1 as a corresponding isotype control (IP IgG1). For mass spectrometric analysis the protein content was visualized by Pierce Coomassie Brilliant Blue G-250 Dye after SDS-PAGE, and the indicated area of interest was extracted for analysis. (**C**) Mass spectrometric analyses of pull-down samples obtained by immunoprecipitation from HEK293, Jurkat, and THP-1 whole cell lysates using the anti-GR (5E4) antibody (AB) and mouse IgG1 as corresponding isotype control (Control). Differential protein abundance compared to isotype control was determined using a two-sample Student’s t test and black circles represent significance with an FDR cut-off of 5%. (**D**) Western blot analysis of GR pulled down from HEK293 whole cell lysates by immunoprecipitation using anti-GR (5E4) antibody Lot #1 (provided by Timea Berki [6]) and Lot #2 (Bio-Rad, Cat# MCA2469, RRID:AB_10844347). The protein content was visualized by incubation with anti-GR (5E4) antibody followed by HRP-conjugated anti-mouse IgG antibody as a secondary reagent. (**E**) Mass spectrometric analyses of pull-down samples obtained by immunoprecipitation from HEK293 whole cell lysates using anti-GR antibodies (AB), 5E4 (Lot #1 and #2), G-5, and pAb PA1, respectively, as well as corresponding isotype controls (Control). Differential protein abundance compared to isotype control was determined using two-sample Student’s t test, and black circles represent significance with an FDR cut-off of 5%. (**F**) Western blot analysis of pull-down samples from HEK293 whole cell lysates obtained by immunoprecipitation using the anti-GR (5E4) antibody. The protein content was visualized by incubation with primary antibodies directed against AMPD2 (Cat# PA5-26127, biotinylated) and TRIM28 (Cat# PA5-27648), respectively, followed by HRP-conjugated streptavidin and anti-rabbit IgG antibody as secondary reagents.

**Figure 3 ijms-23-05049-f003:**
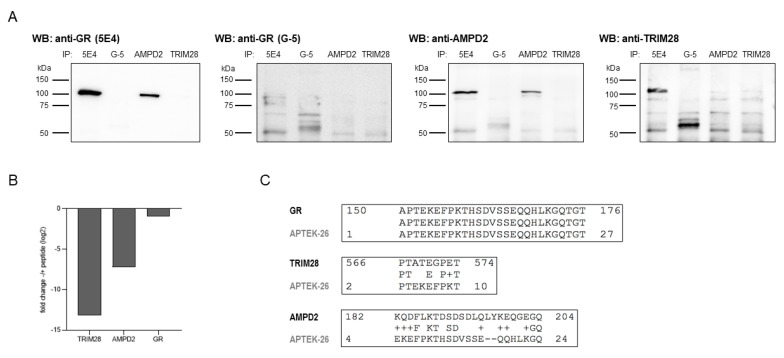
Re-evaluation of anti-GR (5E4) antibody specificity. (**A**) Comparison of target proteins by western blot analysis. Pull-down samples from HEK293 whole cell lysates were obtained by immunoprecipitation using the following antibodies: anti-GR (5E4), anti-GR (G-5), anti-AMPD2 (QQ13), and anti-TRIM28 (Cat# PA5-27648). The protein content was visualized by incubation with primary antibodies directed against GR (5E4, biotinylated), GR (G-5, biotinylated), AMPD2 (PA5-26127, biotinylated), and TRIM28 (Cat# PA5-27648) as indicated, followed by HRP-conjugated streptavidin and anti-rabbit IgG antibody as secondary reagents. (**B**) Mass spectrometric analysis of pull-down samples obtained by IP from HEK293 whole cell lysates using the anti-GR antibody, 5E4, with and without prior two-hour incubation with APTEK-26 peptide. Bar graphs show fold change of peptide incubation to without peptide incubation. (**C**) Amino acid sequences of the newly identified anti-GR (5E4) target proteins, AMPD2 (UniProt ID: Q01433) and TRIM28 (UniProt ID: Q13263), were blasted against the APTEK-26 peptide using NCBI Protein BLAST (https://blast.ncbi.nlm.nih.gov/Blast.cgi, accessed on 14 November 2020) [13].

**Figure 4 ijms-23-05049-f004:**
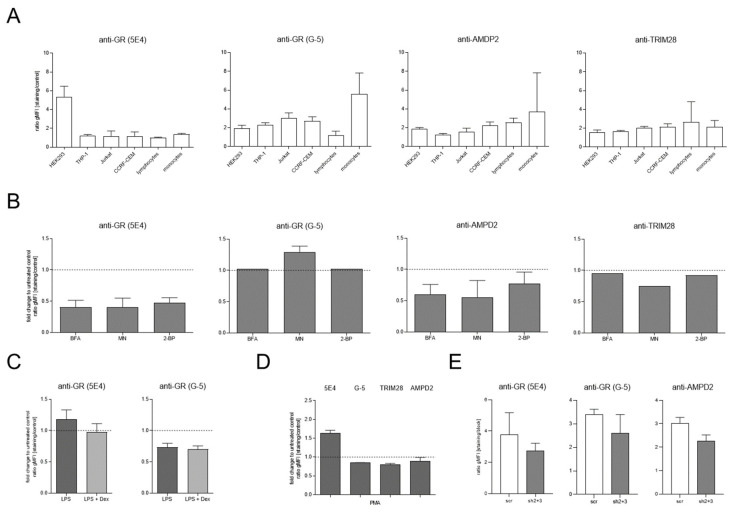
Independent validation of anti-GR (5E4) antibody specificity. (**A**) Multiple cell lines and primary human immune cells were cross-evaluated by flow cytometry. Surface stainings were performed using antibodies directed against GR, AMPD2, and TRIM28, as indicated (n = 3–32). The cells were gated according to Appendix A for analysis. r gMFI represents the ratio of geometric mean fluorescence intensity of staining to unstained control. (**B**) Surface expression of GR, AMPD2, and TRIM28 on HEK293 cells after inhibition of Golgi transport and protein palmitoylation. HEK293 cells were incubated with 1 μg/mL BFA, 0.5 µg/mL MN, and 100 µg/mL 2 BP, respectively, for 24 h, and surface expression was measured by flow cytometry (n = 1–9). The cells were gated according to Appendix A for analysis. r gMFI represents the ratio of geometric mean fluorescence intensity of staining to unstained control. Modification by inhibition of Golgi transport and protein palmitoylation is depicted in relation to untreated control samples. (**C**) Surface expression of the GR was evaluated by flow cytometry using antibody clones, 5E4 and G-5, after immunostimulation. CD14+ monocytes were isolated by magnetic cell separation and incubated with 1 μg/mL LPS ± 10-5 M Dex for 24 h, and surface expression was measured by flow cytometry (n = 2). The cells were gated according to Appendix A for analysis. The data are depicted in relation to untreated control samples. (**D**) Surface expressions of GR, AMPD2, and TRIM28 were evaluated by flow cytometry after immunostimulation. THP–1 cells were incubated with 10 ng/mL PMA for 24 h, and surface expression was measured by flow cytometry (n = 2–8). The cells were gated according to Appendix A for analysis. Modification by immunostimulation is depicted in relation to untreated control samples. (**E**) Surface expression of GR and AMPD2 after GR knockdown was evaluated by flow cytometry (n = 2–3). HEK293 cells stably expressed the indicated shRNA constructs. The cells were gated according to Appendix A for analysis. r gMFI represents the ratio of geometric mean fluorescence intensity of staining to unstained control. Bar graphs depict median and interquartile range. Legend: 2-BP, 2-bromohexadecanoic acid; BFA, brefeldin A; Dex, dexamethasone; LPS, lipopolysaccharide; MN, monensin; scr, scrambled.

**Table 1 ijms-23-05049-t001:** shRNA sequences targeting the GR.

sh1	AAGCTTTCCTGGAGCAAATAT
sh2	CAGACTCAACTTGGAGGATCA
sh3	CTGCATGTACGACCAATGTAA

## Data Availability

Not applicable.

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
