# Peer review of "The Anti-Glucocorticoid Receptor Antibody Clone 5E4: Raising Awareness of Unspecific Antibody Binding"

_ijms, 2022, doi:10.3390/ijms23095049_

Round 1

Reviewer 1 Report

This is an extremely technical work showing that a monoclonal antibody originally identified for being specific for the glucocorticoid receptor (GR) indeed reacted better with two other proteins than with GR.

The data are well described and in general the paper is good. I wonder if this article would be better published in a journal like mAbs.

Here are my more detailed comments.

Line 144:  it should be …by repeating ..

Line 215: RNA interference. But what is the sequence of siRNA? It is not said if it corresponds to the one of the immunizing peptide; in this case, it would be expected to inhibit also AMPD2 expression.

Line 427: shouldn’t it be: … blocking of the staining with 100-fold excess UNCONJUGATED antibody

The legends on Fig 2C and E are unreadable. Characters must be increased in size.

Legends Fig 3. It would be better to write in a more readable way about the IP. IP could be written only once on the left of the Fig and so the different precipitating Ab could be in greater characters. Also, I would suggest to write in front of the antibodies used for western blot: WB:Ab.

Legends Fig 4. are nearly unreadable. Characters must be increased in size.

The finding that AMPD2 seems to be the preferred target of the 5E4 mAb, but TRIM28 is better competed by the APTEK-26 peptide in Immunoprecipitation should be discussed.

Author Response

This is an extremely technical work showing that a monoclonal antibody originally identified for being specific for the glucocorticoid receptor (GR) indeed reacted better with two other proteins than with GR.

The data are well described and in general the paper is good. I wonder if this article would be better published in a journal like mAbs.

We thank the reviewer for his/her positive evaluation of our manuscript. With respect to the suggested journal, we acknowledge its suitability. However, we are convinced that our manuscript would perfectly complement the special issue “Advances in Antibody Design and Antigenic Peptide Targeting 2.0” by highlighting the importance of rigorous antibody validation.

Line 144:  it should be …by repeating ..

We thank the reviewer for pointing this out. We meant to describe replicate experiments and have adjusted the wording accordingly.

“[…] First, we investigated potential antibody batch effects by replicate experiments applying the antibody clone 5E4 provided by different manufacturers. […]”

Line 215: RNA interference. But what is the sequence of siRNA? It is not said if it corresponds to the one of the immunizing peptide; in this case, it would be expected to inhibit also AMPD2 expression.

The shRNA sequences are provided in the Methods section in Table 1. The immunizing peptide is encoded by nucleotides 667-747 (NM_001364180.2) of the NR3C1 gene. The shRNA sequences, on the other hand, localize to the following positions of the gene sequence: 1131-1151 (sh1), 1877-1897 (sh2), and 2130-2150 (sh3). Besides, similarities in the amino acid sequences of the immunizing peptide APTEK-26 and AMPD2 do not necessarily suggest an underlying overlap in the genetic sequence due to the greater variation potential of the genetic code.

However, we also blasted the AMPD2 gene sequence (NM_001368809.2) with the three respective shRNA sequences and did indeed identify some overlap (max. 9 nucleotides). While these similarities are independent of the immunizing peptide, they might however account for a reduction in AMPD2 protein expression observed in cells transduced with GR-targeting shRNA. These off-target effects are not uncommon and have to be taken into account when performing knockdown by RNA interference. This aspect is now addressed in the Discussion section.

“[…] Additionally, alignment of up to 9 nucleotides was identified when blasting GR-targeting shRNA (Table 1) and AMPD2 gene sequences, accounting for potential off-target effects. […]”

Line 427: shouldn’t it be: … blocking of the staining with 100-fold excess UNCONJUGATED antibody

We thank the reviewer for highlighting this mistake. It has been corrected.

“[…] Successful blocking of the staining with 100-fold excess unconjugated antibody excluded the possibility of unspecific antibody attachment to the cell surface. […]”

The legends on Fig 2C and E are unreadable. Characters must be increased in size.

Legends Fig 3. It would be better to write in a more readable way about the IP. IP could be written only once on the left of the Fig and so the different precipitating Ab could be in greater characters. Also, I would suggest to write in front of the antibodies used for western blot: WB:Ab. Legends Fig 4. are nearly unreadable. Characters must be increased in size.

We apologize for the poor quality of the figures in the document and thank the reviewer for his/her helpful suggestions for improvement. The figure legends were enlarged and adjusted accordingly. Also, a high-resolution version of each figure was provided in pdf format upon first submission. This quality was unfortunately not reflected in the word document. We apologize for the inconvenience.

The finding that AMPD2 seems to be the preferred target of the 5E4 mAb, but TRIM28 is better competed by the APTEK-26 peptide in Immunoprecipitation should be discussed.

We thank the reviewer for raising this important aspect. The finding is now discussed in the manuscript.

“[…] Interestingly, while IP-MS identified AMPD2 as the target protein most efficiently pulled down by the anti-GR (5E4) antibody, pre-incubation with the immunizing peptide APTEK-26 had a more pronounced effect on the IP of TRIM28. These findings suggest that the area relevant for AMPD2 binding might extend beyond the APTEK-26 binding site. Also, different binding affinities might play a role in this context. These characteristics were not determined as part of our work. […]”

Reviewer 2 Report

The paper reported by Timo Gaber et al. describes an example of relevant unspecific binding of a widely commercially available monoclonal antibody.   Through IP-MS experiments the authors identified TRIM28 and AMPD2 as the predominant targets of the anti-GR antibody 5E4.   The result by western blot analysis also confirmed these results.   The study reported in this paper promotes rigorous antibody validation by underlining the dangers of misleading results produced by the less stringent method.  The results reported in this paper are of great interest to researchers in the field of protein science as well as antibody engineering.   Thus, I recommend the publication in the International Journal of Molecular Sciences.   Aside from the recommendation, the flowing points should be cleared before publication.

1) Although the author describes the unspecific binding of antibody 5E4 with TRIM28 and AMPD2 by several experiments. However, the binding affinities (or Kds) of 5E4 with TRIM28 and AMPD2 were not given in the paper. These physical values were very important to show the unspecific reactivity of 5E4 with TRIM28 and AMPD2.   The authors should give these values by surface plasmon methods such as BIA-CORE or other methods.

2) Some legends in Figures 1-4 are very small. It was very hard to read them. Please enlarge all of them.

Author Response

The paper reported by Timo Gaber et al. describes an example of relevant unspecific binding of a widely commercially available monoclonal antibody.   Through IP-MS experiments the authors identified TRIM28 and AMPD2 as the predominant targets of the anti-GR antibody 5E4.   The result by western blot analysis also confirmed these results.   The study reported in this paper promotes rigorous antibody validation by underlining the dangers of misleading results produced by the less stringent method.  The results reported in this paper are of great interest to researchers in the field of protein science as well as antibody engineering.   Thus, I recommend the publication in the International Journal of Molecular Sciences.   Aside from the recommendation, the flowing points should be cleared before publication.

We are grateful to reviewer 2 for his/her positive assessment of our manuscript.

1) Although the author describes the unspecific binding of antibody 5E4 with TRIM28 and AMPD2 by several experiments. However, the binding affinities (or Kds) of 5E4 with TRIM28 and AMPD2 were not given in the paper. These physical values were very important to show the unspecific reactivity of 5E4 with TRIM28 and AMPD2.   The authors should give these values by surface plasmon methods such as BIA-CORE or other methods.

We agree with the reviewer that the determination of the binding affinities of the anti-GR antibody clone 5E4 would be technically very interesting and a valuable addition to the data set presented in our manuscript. The technique suggested by the reviewer would be suitable to achieve these results and help delineate the characteristics of the described antibody in even more detail. However, we are of the opinion that this experiment is beyond the scope of our manuscript and would not be feasible in the 7-day revision period. The principle message of the unspecific binding of the monoclonal antibody is in our opinion valid without these additional experiments. As we acknowledge the value of this suggestion, we have however added this limitation to the discussion of our results.

“[…] Interestingly, while IP-MS identified AMPD2 as the target protein most efficiently pulled down by the anti-GR (5E4) antibody, pre-incubation with the immunizing peptide APTEK-26 had a more pronounced effect on the IP of TRIM28. These findings suggest that the area relevant for AMPD2 binding might extend beyond the APTEK-26 binding site. Also, different binding affinities might play a role in this context. These characteristics were not determined as part of our work. […]”

2) Some legends in Figures 1-4 are very small. It was very hard to read them. Please enlarge all of them.

We apologize for the poor quality of the figures in the document. The figure legends were enlarged accordingly. Also, a high-resolution version of each figure was provided in pdf format upon first submission. This quality was unfortunately not reflected in the word document. We apologize for the inconvenience.

Round 2

Reviewer 2 Report

The revisions made by the atuthors are satisfactory.   Now, I recommend the publication in IJMS.